# Epithelial-to-Mesenchymal Transition Enhances Cancer Cell Sensitivity to Cytotoxic Effects of Cold Atmospheric Plasmas in Breast and Bladder Cancer Systems

**DOI:** 10.3390/cancers13122889

**Published:** 2021-06-09

**Authors:** Peiyu Wang, Renwu Zhou, Patrick Thomas, Liqian Zhao, Rusen Zhou, Susmita Mandal, Mohit Kumar Jolly, Derek J. Richard, Bernd H. A. Rehm, Kostya (Ken) Ostrikov, Xiaofeng Dai, Elizabeth D. Williams, Erik W. Thompson

**Affiliations:** 1Wuxi School of Medicine, Jiangnan University, Wuxi 214122, China; 2Queensland University of Technology (QUT), School of Biomedical Sciences, Brisbane 4059, Australia; p37.wang@hdr.qut.edu.au (P.W.); pb.thomas@qut.edu.au (P.T.);; 3Translational Research Institute, Woolloongabba, Queensland 4102, Australia; 4School of Chemical and Biomolecular Engineering, The University of Sydney, Sydney 2006, Australia; renwu.zhou@sydney.edu.au (R.Z.);; 5Queensland Bladder Cancer Initiative (QBCI), Woolloongabba, Queensland 4102, Australia; 6The First School of Clinical Medicine, Southern Medical University, Guangzhou 510515, China; 7Centre for BioSystems Science and Engineering, Indian Institute of Science, Bangalore 560012, India; 8Centre for Cell Factories and Biopolymers, Griffith Institute for Drug Discovery, Griffith University, Nathan, Queensland 4111, Australia; 9School of Chemistry and Physics, Queensland University of Technology, Brisbane 4000, Australia

**Keywords:** cold atmospheric plasma (CAP), plasma-activated medium (PAM), epithelial–mesenchymal transition (EMT), reactive oxygen species (ROS)

## Abstract

**Simple Summary:**

Cold atmospheric plasma (CAP) and plasma-activated medium (PAM) are known to selectively kill cancer cells, however the efficacy of CAP in cancer cells following epithelial-mesenchymal transition (EMT), a process which endows cancer cells with increased stemness, metastatic potential, and resistance to conventional therapies, has not been previously examined. We have used several established models of EMT to show that PAM is significantly more active in cancer cells exhibiting EMT than their epithelial counterparts. We further show that this enhancement correlated with increased levels of reactive oxygen species (ROS) in the mesenchymally-shifted cell lines.

**Abstract:**

Cold atmospheric plasma (CAP) has emerged as a highly selective anticancer agent, most recently in the form of plasma-activated medium (PAM). Since epithelial–mesenchymal transition (EMT) has been implicated in resistance to various cancer therapies, we assessed whether EMT status is associated with PAM response. Mesenchymal breast cancer cell lines, as well as the mesenchymal variant in an isogenic EMT/MET human breast cancer cell system (PMC42-ET/LA), were more sensitive to PAM treatment than their epithelial counterparts, contrary to their responses to other therapies. The same trend was seen in luminal muscle-invasive bladder cancer model (TSU-Pr1/B1/B2) and the non-muscle-invasive basal 5637 bladder cancer cell line. Three-dimensional spheroid cultures of the bladder cancer cell lines were less sensitive to the PAM treatment compared to their two-dimensional counterparts; however, incrementally better responses were again seen in more mesenchymally-shifted cell lines. This study provides evidence that PAM preferentially inhibits mesenchymally-shifted carcinoma cells, which have been associated with resistance to other therapies. Thus, PAM may represent a novel treatment that can selectively inhibit triple-negative breast cancers and a subset of aggressive bladder cancers, which tend to be more mesenchymal. Our approach may potentially be utilized for other aggressive cancers exhibiting EMT and opens new opportunities for CAP and PAM as a promising new onco-therapy.

## 1. Introduction

Cold atmospheric plasma (CAP) represents an emerging onco-therapeutic approach offering a new opportunity to effectively manage aggressive cancers [1,2]. CAP is generated under atmospheric pressure conditions at room temperature, and is a source of charged particles, free radicals, metastable species, UV radiation and electric fields [3]. In addition to the direct CAP strategy, CAP activity can also be transferred to liquids (i.e., plasma activated medium (PAM)), adding flexibility to the application of plasma therapy [4,5,6]. PAM, produced by activating culture medium with CAP, has been found to selectively cause a significant reduction in the viability of cancer cells, including breast cancer [7], bladder cancer [8], osteosarcoma [9], and skin cancer [10] cells.

Epithelial-mesenchymal transition (EMT) is an important cellular process during organ development through which cells lose their epithelial features at the cellular and molecular levels, and acquire mesenchymal traits such as stellate morphology, reduced adhesion, and increased migration [11]. EMT processes are hijacked by malignant epithelial cells, leading to increased mobility and invasion, and in some cases, reduced proliferation [12]. Successful metastasis also requires MET to revert the phenotype to an epithelial state [13].

We used several EMT model systems available in breast and bladder cancer to explore the relationship between EMT status and CAP response in this study. We and others have previously shown that intrinsic molecular subgroups of human breast cancer cell lines differed in terms of EMT state, with the most aggressive Basal B/Mesenchymal subgroup being more mesenchymal than the Basal/Basal A, which in turn was more mesenchymal that the Luminal subgroup [14,15,16,17]. Some of these cell lines have also been aligned with newly defined clinical subgroups of triple negative breast cancers (TNBCs) [18,19,20]. We have examined the PAM responses of MCF-7 (Luminal and ER/PR+), MDA-MB-468 (Basal A/ Basal-Like 1 (BL1), more epithelial and TNBC) and MDA-MB-231 (Basal B / Mesenchymal (M)–more mesenchymal and TNBC), as well as the isogenic breast cancer PMC42-ET/LA model, in which the epithelial PMC42-LA cell line arose spontaneously from the parental, mesenchymal PMC42-ET [21,22,23,24], although both map tentatively to the Basal B intrinsic classifier [24]. We also assessed bladder cancer cell lines that represent non-muscle invasive bladder cancer (NMIBC; 5637 cells derived from basal bladder cancer and relatively epithelial [25]) and muscle invasive bladder cancer (MIBC; T24/TSU-Pr1 cells derived from high grade transitional cell carcinoma [2]), as well as TSU-Pr1-derived sublines (TSU-Pr1-B1, -B2). The TSU-Pr1-derived B1 and B2 sublines were selected for increased metastatic colonization of mouse bones and exhibit sequential increases in epithelial phenotype [26,27]. Collectively these models provide an opportunity to assess the relationship between EMT and PAM response in two different tumor streams.

## 2. Materials and Methods

### 2.1. Reagents

CellROX^®^ Orange (C10443, Thermo Fisher Scientific, San Diego, CA, USA), Hoechst 33342 (H3570, Themo Fisher Scientific), Happy Cell Advanced Suspension Matrix (HC-ASM; Vale Life Sciences, Biocroi, Ireland), CellEvent™ Caspase-3/7 Green ReadyProbes™ Reagent (C10423, Thermo Fisher Scientific), Cell CellTiter-Glo^®^ 3D Cell Viability Assay (Promega, Madison, WI, USA)

### 2.2. PAM Generation

The CAP device used in this study was a typical commercial argon plasma jet (model kINPen 09, INP Greifswald, Germany) [28]. During the experiment, pure argon gas was kept at the flow rate of 5.0 Standard Liter per Minute. Aliquots of 1.5 mL medium containing serum in 2 mL centrifuge tubes were activated by CAP for 10 min and named ‘10PAM’ here. The distance between the plasma jet nozzle to the medium surface was 10 mm. PAM was then diluted to different concentrations, designated by the percent remaining (e.g., 70% 10PAM refers to 70% concentration at use). After treatment, different concentrations of PAM were used in cultured cells, which were seeded in 96-well plates.

### 2.3. Cell Culture

Human breast cancer cell lines MCF-7, MDA-MB-468, and MDA-MB-231, originally from the American Type Culture Collection (ATCC; Manassas, VA, USA) were transferred from the Lombardi Cancer Center, USA. Parental PMC42-ET cells and the spontaneously derived, epithelially shifted PMC42-LA cells were previously described [24]. Human bladder cancer cell line 5637 (ATCC^®^HTB-9™) was purchased from the ATCC. The T24/TSU-Pr1 cells are as previously described [26,27]. Cell lines were cultured in Dulbecco’s modified Eagle’s medium (DMEM, Gibco, Thermo Fisher Scientific, Melbourne, VIC, AU) supplemented with 10% foetal bovine serum (FBS, Gibco Thermo Fisher Scientific, AU) (MCF-7, MDA-MB-468, MDA-MB-231 and TSU-Pr1/B1/B2), or Roswell Park Memorial Institute (RPMI)-1640 (Gibco Thermo Fisher Scientific AU) with 10% FBS (PMC42-ET/LA, 5637). All cell lines were maintained at 37 °C in a humidified incubator (Panasonic, Osaka, Japan) containing 5% CO_2_; cells were grown in tissue culture flasks (Corning, NY, USA) and passaged routinely with trypsin (Sigma-Aldrich, St. Louis, MO, USA) upon reaching 70% confluence. STR profiling and regular Mycoplasma testing was performed (Universal Mycoplasma Detection kit; ATCC).

### 2.4. EMT Score Analysis

EMT scores were derived from RNA expression data as previously described [29,30,31] using 3 different approaches. The MLR (multinomial logistic regression) method [30] is based on an iterative algorithm performing probabilistic categorisation of cell lines into epithelial (E), hybrid and mesenchymal (M). The 76GS method calculates scores based on weighted average of gene expression values for 76 genes pertinent to EMT [31]. The KS method is based on Kolmogorov-Smirnov test using E vs. M gene lists identified separately for cell lines and for tumors [29]. We have earlier shown a good concordance among the three methods across various datasets [32]. Publicly available RNA-Seq data from CCLE (MCF-7, MDA-MB-468, MDA-MB-231, 5637, T24; [33]), or NCI-60 (BR-MCF-7, BR-MDA-MB-231; [34]) were accessed and subjected to comparative EMT score analyses. Normalized PMC42-LA and ET RNA-Seq data [35] was used for the PMC42 system; the raw data is available at https://submit.ncbi.nlm.nih.gov/subs/bioproject/SUB1515288/overview (accessed on 23 May 2016).

### 2.5. Live and Dead Cell Viability

Cells were trypsinized, centrifuged (1000× *g*, 5 min, room temperature (RT)), resuspended, and seeded in 96-well plates at 5000 cells per well. The indicated treatments commenced 24 h post-seeding. At the experimental endpoints, cellular supernatants were removed and cells were washed with phosphate-buffered saline (PBS). PBS (50 μL) containing propidium iodide (10 μg/mL) and Hoechst 33342 (10 μg/mL) was added to each well for 30 min, and the plate was incubated at 37 °C. Cells were scanned using an InCell Analyzer 6500HS (GE Healthcare, Chicago, IL, USA, 10× objective). For imaging propidium iodide (PI) staining, a 642 nm laser was used for excitation (solid-state laser rated power = 30 mW) together with a far-red 650–720 nm emission filter. For imaging Hoechst 33342 staining, a 405 nm laser was used for excitation (solid-state laser rated power = 100 mW) together with a blue 430–470 nm emission filter. Live/dead cell analysis was performed using IN Carta analysis software128 (GE Healthcare). The analysis parameter “settings” for the Hoechst 33342 channel were “Fast” for segmentation model, “Nuclei” for target type, “40 μm^2^” for min. target area, “60/100” for sensitivity, “40–500 μm^2^” for area filters and “3000–30,000 a.u.” for intensity filters. The analysis parameter settings for the PI channel were “robust” for segmentation model, “20 μm” for diameter, “94/100” for sensitivity, “40–500 μm^2^” for area filters and “80–10,000 a.u.” for intensity filters. Cell viability = cell count (PI)/cell count (Hoechst 33342) × 100%.

### 2.6. ROS Detection

The cells for experiments were trypsinized, centrifuged (1000× *g*, 5 min, RT), resuspended, and seeded in 96-well plates at 5000 cells per well. The indicated treatments commenced 24 h post-seeding, and cells were incubated with treatments for the specified time duration. Following supernatant removal, PBS containing 5 μM CellROX Orange Reagent was added to cells and incubated for 30 min at 37 °C. Medium was removed and cells were washed 3 times with PBS after which the cells were stained with 10 μg/mL Hoechst 33342 in PBS for 30 min at 37 °C. Cells were scanned using an InCell Analyzer 6500HS (60× objective, excitation 561 nm solid-state laser rated power = 30 mW, emission filter Orange 570–720 nm). ROS level qualification was performed using IN Carta analysis software 128. The analysis for Hoechst 33342 channel was as described in Section 2.5. The analysis parameter “settings” for the CellROX Orange channel was “Fast Puncta” for segmentation model, “Organelles” for target type, “1 μm^2^” for min. target area, “80/100” for sensitivity, “0–50 μm^2^” for area filters and “200–10,000 a.u.” for intensity filters. 

Relative ROS value in cell = the ROS intensity mean value (treated group)/the ROS intensity mean value (untreated group).

### 2.7. Spheroid Generation

Following growth in appropriate complete medium and trypsinization for 5 min at 37 °C, bladder cancer cells were washed twice in PBS. Single cell suspensions were centrifuged at 133× *g* for 5 min at RT, and the supernatant discarded. Cell pellets were resuspended in 50 mL Bioreactor tubes (Vale Life Sciences, Brisbane, Australia) in pre-warmed 2× Happy Cell Advanced Scientific Media (HC-ASM; Vale Life Sciences, Brisbane, QLD, Australia) and complete growth medium to a density of 100,000 cells/mL and placed in an incubator at 37 °C, 5% CO_2_ for daily imaging. Images of developing spheroids were taken under the Eclipse Ti2-E inverted phase-contrast microscope (Nikon Instruments Europe B.V., Amsterdam, The Netherlands; 20×/0.75 numerical aperture (NA), differential interference contrast (DIC) optics) equipped with a digital camera. Spheroids formed 72 h after seeding were used in experiments.

### 2.8. Spheroid Viability Assays and Growth Kinetics

Spheroid cultures were separated into equal volumes and centrifuged at 150× *g* for 5 min at RT. Spheroids were resuspended in HC-ASM media containing PBS, PAM, or cisplatin (10 µM), and approximately 10–20 intact spheroids were dispensed per well into 96-well ultra-flat-bottom ultra-low-attachment plates (Corning). Spheroids were returned to the incubator (37 °C, 5% CO_2_) for 6 days. Following this period, the CellTiter-Glo 3D Cell Viability Assay (Promega, Madison, WI, USA) was undertaken with luminescence measured on the FLUOstar OPTIMA (BMG Labtech, Ortenberg, Germany), according to manufacturer’s protocol. Mean spheroid area (µM^2^) (total number of pixels used to image the surface area covered by the cell) was determined following treatment using NIS-Elements Advanced Research. For apoptosis analysis, 100 µL of CellEvent drug solution (Thermo Fisher Scientific) and Hoechst 3342 (10 µg/mL) were added to each well as per manufacturer’s instructions. Images were obtained as described above using excitation and emission filters (GFP Ex 465/95, Em 500–550 nm; DAPI Ex 340/80, Em 430–470 nm) on a Nikon Eclipse Ti2-E inverted phase-contrast microscope imaging system to visualize caspase-3/7 and cell nuclei.

### 2.9. RNA Extraction, cDNA Synthesis, and Realtime Quantitative PCR (RT-qPCR) 

Cells were seeded in a 48-well plate overnight and treated by indicated methods. Cells were incubated with 100 μL TRIzol (Life Technologies, New York, NY, USA) on ice for 5 min. Subsequently, the suspension was transferred to 2 mL tubes and 20 μL chloroform was added with 15 s vigorous shaking by hand. After 3 min RT incubation, tubes were centrifuged (13,000× *g*, 15 min, 4 °C). The colorless upper phase was carefully transferred to Bioline Isolate II RNA Micro filters (Bioline, Sydney, NSW, Australia), and the kit manufacturer’s instructions were followed. The RNA was diluted with 10 μL RNase-free water. The RNA concentration of each group was quantified and normalized to 0.5 μg/μL. The synthesis of cDNA was performed as per instructions from the SensiFASTTM cDNA Synthesis kit (Bioline). The cDNA and targeted gene primers (Table 1) were mixed with dNTPs, buffer, and SYBR Green Master Mix (Thermo Fisher Scientific, AU) in a 386-well plate at 10 μL per well as protocol indicated. Forty cycles of real-time qPCR were detected by a ViiA7 system (Applied Biosystems, Foster, CA, USA) and analyzed by QuantstudioTM Real-Time PCR software v1.1 (Applied Biosystems, Life Technologies, Thornton, NSW, Australia). Raw ϪCt = Ct(L32)–Ct (target gene)

### 2.10. Statistics

Statistical tests were performed using GraphPad Prism 8.4.3. The column analysis *t* test (and nonparametric tests) was used to compare different treatment groups. Mean ± standard error of the mean (SEM) is plotted. “n.s.” indicates *p* > 0.05 (defined as not significant), “*” indicates *p* < 0.05, “**” indicates *p* < 0.01, “***” indicates *p* < 0.001, and “****” indicates *p* < 0.0001.

## 3. Results

### 3.1. Mesenchymal Breast and Bladder Cell Lines Are More Sensitive to PAM Than Their Epithelial Counterparts in 2D Culture

We tested the relationship between EMT status and viability of cells in response to PAM treatment using three EMT models (Figure 1a,c). In human breast cancer cell lines, we observed incrementally greater responses of the cell lines to PAM treatment as they progressed from Luminal (MCF-7) to Basal A/BL1 (MDA-MB-468) and Basal B/M (MDA-MB-231). At the same PAM concentration, taking 100% 10PAM as an example, the cell viability of MCF-7 was 62.99 ± 2.52%, while that of MDA-MB-468 was 40.29 ± 6% and that of MDA-MB-231 was 16.02 ± 5.02%. Consistent with this, the parental mesenchymal-like PMC42-ET cells showed a greater response to PAM than the epithelially-shifted, spontaneous derivative, PMC42-LA. Among the bladder cancer cell lines, the NMIBC-derived, relatively epithelial 5637 cells showed lower responses to PAM than the MIBC-derived mesenchymal TSU-Pr1. Furthermore, the epithelially-shifted variants TSU-Pr1-B1 and -B2 showed reduced PAM responsiveness as compared to the parental cells. This was most evident in the TSU-Pr1-B2 derivative, which is shifted further along the epithelial axis than the TSU-Pr1-B1 cells. Therefore, in all three EMT systems, the mesenchymal cell lines were more sensitive to the PAM treatment than their epithelial counterparts. 

A concentration of 70% 10PAM was then chosen for the time gradient experiment (Figure 1b,d), as the cytotoxic effect of 70% 10PAM had been observed in all the cell lines. The results showed that mesenchymal cell lines responded to 70% PAM at earlier time points than the epithelial counterparts in each system. Taking breast cancer cell lines as an example. MCF-7 cells took 9 h to respond to 70% 10PAM, while MDA-MB-231 cells took 6 h. From the results of Figure 1c,d, we could gain the same conclusion. The ET/LA (Figure 1e,f) and bladder cancer cell lines (Figure 1g,h) showed the same patterns.

In terms of the cell number changes (Appendix A), the PAM treatments did not influence the attached cell numbers of MCF-7 and MDA-MB-468 cell lines, but did inhibit MDA-MB-231 cell line, which indicated MDA-MB-231 is more sensitive than the other breast cancer cell lines. Similarly, the mesenchymal cells were more sensitive than the epithelial cell lines in the other EMT models (Appendix A).

### 3.2. ROS Level Increase in Mesenchymal Cell Lines Is Higher Than in Epithalial Counterparts Following PAM Treatment

As the active agents of PAM include reactive oxygen species (ROS), we measured ROS levels across these three EMT models. As shown in Figure 2a, the more mesenchymal cell lines/variants had higher ROS levels than the epithelial cell lines in each of the three EMT models. Compared to MCF-7, the resting ROS level was significantly higher by an average of 1.5-fold (*p* = 0.0039) in MDA-MB-468 cells and 2.3-fold in MDA-MB-231 cells (*p* = 0.0013). The ET/LA and bladder cancer cell line systems also showed significant differences, with higher ROS levels seen in the mesenchymal cell lines/variants (ET/LA, 2.6-fold, *p* = 0.0160; TSU-Pr1/B2, 2.7-fold, *p* = 0.0284). Thus, the observed increased PAM sensitivity of more mesenchymal cell lines is likely due to the higher baseline ROS level in these cells.

The ROS levels after 70% 10PAM 12 h treatment increased in all cell lines (Figure 2b), however those in mesenchymal cell lines showed a higher increase than the epithelial cell lines. The fold changes of ROS levels were significantly (*p* = 0.016) lower in PMC42-LA (2.1 ± 0.9) than PMC42-ET (4.9 ± 0.6), and lower in TSU-Pr1-B2 (1.9 ± 0.4) than its’ parental TSU-Pr1 (3.3 ± 0.2) cells (*p* = 0.028) (Figure 2b).

The cell morphologies of epithelial cell lines MCF-7, PMC42-LA, TSU-Pr1-B2, and TSU-Pr1-B1 were cuboidal and closely adherent to each other (Figure 2d), and we observed that these cells had relatively low levels of ROS staining in the absence of PAM treatment (Figure 2a). In contrast, the mesenchymal counterparts (MDA-MB-231, PMC42-ET, 5637, and TSU-Pr1) exhibited a spindle-like and spread-out morphology, as well as appreciably higher ROS levels. After PAM treatment, a proportion of the cells underwent cell death (indicated by the white arrows in Figure 2d). The nuclei of the cells exhibiting PAM-induced apoptosis appeared shrunken or were dispersed into apoptotic bodies. The ROS levels seen in these cells in the mesenchymal cell lines/variants were dramatically higher than those in the epithelial cell lines. Interestingly, almost no ROS were detected in PMC42-LA cells.

### 3.3. Transcriptomally Derived EMT Scores Correlate with PAM Response 

To assess the relationships between EMT status, PAM response, and ROS levels more closely, we derived EMT scores from RNA-Seq data from the cell lines where available, which allowed assessment of all cell lines except the B1 and B2 variants of the TSU-Pr1 (Figure 3). Significant linear correlations were seen between cell viability responses to PAM and all three EMT scores, the strongest being with the 76GS score (Figure 3a; R^2^ = 0.7800, *p* < 0.0001). Strong levels of association were also seen with the two other EMT score algorithms (MLR, Appendix A, R^2^ = 0.4466, *p* < 0.0001; KS, Appendix A, R^2^ = 0.5319, *p* < 0.0001). In all three EMT scoring algorithms, similar results were obtained with public data for different cell lines from different datasets (e.g., CCLE and NCI-60) or between replicate cultures of PMC42-LA and -ET (designated as −1 and −2, respectively, in Figure 3). These results further support the observations that cells with higher levels of EMT score are more amenable to PAM response.

Given the proposed role that ROS play in cancer-selective PAM responses, we also tested the correlations between the ROS levels in untreated (Figure 3b for 76GS) or PAM-treated (Figure 3c for 76GS) cultures. Before PAM treatment, the R^2^ value was only 0.1251 (76GS), classified as a very weak linear correlation, and even lower values were seen in MLR (Appendix A, R^2^ = 0.2546) and KS (Appendix A, R^2^ = 0.0751). The PAM treatment increased the 76GS correlation value (R^2^ = 0.5169, Figure 3c), supporting the essential role attributed to ROS in mesenchymal cell sensitivity to PAM. As with the levels in untreated cultures, lesser values were seen using the MLR (Appendix A, R^2^ = 0.4248, *p* < 0.0001) and KS (Appendix A, R^2^ = 0. 3860, *p* = 0.0001). We also assessed the potential association between the effect (fold change) of PAM treatment on ROS (Appendix A), which did show weak but significant associations for 76GS (R^2^ = 0.5493 *p* < 0.0001) and KS (R^2^ = 0.2082, *p* = 0.0076), but not for MLR (R^2^ = 0.07396, NS).

### 3.4. Serum-Containing Media Counteracts the Inhibition of Cell Viability Caused by PAM in BREASt and Bladder Cancer Cell Lines

Fetal bovine serum (FBS) is a complex mixture of multiple factors and is generally regarded as an antioxidant reagent, including ascorbic acid, alpha-tocopherol, beta-caro-tene, glutathione (GSH), uric acid, and bilirubin [36]. There has been controversy over the influence of serum on the effect of PAM in killing cancer cells [37]. PAM-induced apoptosis has been previously shown to be reduced with increased FBS concentration; however, the relationship of these effects with EMT status remains unknown. We were motivated to test the effects of serum on PAM response in our EMT models. FBS was found to rescue the PAM-induced loss of cell viability (Figure 4). Taking the breast cancer cell lines as an example, 10% FBS rescued 19.5 ± 7.3% (*p* = 0.027) of cell viability in MCF-7, 33.6 ± 2.2% (*p* = 0.0030) in MDA-MB-468, and 41.3 ± 4.3% (*p* = 0.0007) in MDA-MB-231 cells. The presence of serum also rescued 26.0 ± 7.5% (*p* = 0.0356) of cell viability in PMC42-LA and 33.98 ± 7.1% (*p* = 0.0192) of cell viability in PMC42-ET cells. Interestingly, both the margin of cell viability with and without serum and the p value advanced incrementally along the EMT axis in the mesenchymal direction in breast cancer cell lines.

In the bladder cancer cell lines, the presence of serum rescued the PAM effects on cell viability by 20.2 ± 5.7% (*p* = 0.0047) in the most mesenchymal TSU-Pr1 cell line, by 29.4 ± 7.5% (*p* = 0.0489) in the TSU-Pr1-B1 cell line, and by 33.0 ± 9.6% (*p* = 0.0419) in the TSU-Pr1-B2 cell line. Although the ability of serum to preserve cell viability is not the strongest in TSU-Pr1, the effect is the most consistent. These results may also reflect that this cell line is the most sensitive among all the lines tested to PAM treatment, even in the presence of serum.

As a control for the high protein content in FBS, we also tested bovine serum albumin (BSA). BSA (10%) was added to the different concentrations of PAM as we did for FBS. The results are shown in Appendix A–c. Surprisingly, unlike FBS, the BSA protein did not protect the cells but actually exacerbated the PAM-induced cell viability decline. 

We also tested the direct effects of PAM treatment of FBS for 10 min, as for PAM, prior to addition to the cultures. FBS activated by plasma (plasma-activated serum (PASerum)) also induced cell death in MDA-MB-468 and MDA-MB-231 cultures after 12 h treatment compared to the same treatment time with the equivalent amount of PAM without FBS. There were no significant differences between 100% 10PAM and 100% 10PASerum in MCF-7 and MDA-MB-468 cell lines; however, MDA-MB-231 cells were more sensitive to PAM treatment than PASerum (Appendix A).

### 3.5. PAM Effects on EMT Marker Expression Required the Presence of FBS in MDA-MB-231 Cells

Since the results above showed that serum could attenuate the PAM-induced inhibition of cell viability, we were motivated to test the effects of PAM on EMT marker gene expression in 10% FBS or in a serum-free environment. The mRNA expression level of epithelial marker (CDH1, also known as e-cadherin), mesenchymal markers (CDH2, EpCAM, VIM, SNAI1, SNAI2, ZEB1), and stemness markers (CD24, CD44) were detected via real-time qPCR technology. The mesenchymal MDA-MB-231 breast cancer cell line had lower mRNA expression of epithelial marker CDH1 and higher expression of mesenchymal markers VIM and ZEB1 (Figure 5) compared to the MCF-7 and MDA-MB-468 cell lines (Appendix A–d). The other markers were not altered significantly. Treatment with 30% 10PAM for 3 days in the absence of FBS induced significant downregulation in the levels of the mesenchymal markers CDH2 (*p* = 0.0090) and VIM (*p* = 0.0179), while the epithelial marker EpCAM was upregulated (*p* = 0.0258); other markers were not changed. Although the levels of CDH2, VIM, and EpCAM also changed in the presence of serum, the degree of change appeared much slighter than observed under serum-free conditions; however, these experiments were performed separately and cannot be directly compared. After 30% 10PAM treatment of MDA-MB-231 cells in the presence of 10% FBS for 3 days, levels of almost all markers underwent significant changes, except EpCAM. Again, epithelial markers CDH1 (*p* = 0.0010) and CD24 (*p* = 0.0002) were upregulated, while mesenchymal markers (CDH2, *p* = 0.0131; VIM, *p* = 0.0055; SNAI1, *p* = 0.0029; SNAI2, *p* = 0.0011; Zeb1, *p* = 0.0005) and the stem cell marker CD44 (*p* = 0.0005) were downregulated. These results indicated that the cells had undergone the MET process after 3 days of 10% FBS 30% 10PAM treatment, becoming more epithelial. Since the cells were less responsive to PAM in the presence of 10% FBS, these results are consistent with the conclusion drawn from the data shown in Figure 1 that epithelial cells are more resistant to PAM treatment. We can also conclude that serum may play a critical role in PAM-induced EMT mRNA marker changes in the MDA-MB-231 cell line. However, these results were not consistently seen in the other two cell lines (Appendix A). MCF-7 cells showed only non-significant changes for PAM treatment, regardless of serum presence, although these cells are considerably less responsive to PAM in general. MDA-MB-468 cells showed inconsistent EMT marker changes only in presence of FBS, with both epithelial (CDH1, EpCAM) and mesenchymal (VIM) markers being downregulated, and SNAI1 being upregulated.

### 3.6. The 3D Bladder Spheroid Model Confirms the Association of the Mesenchymal State and Cell Sensitivity to PAM 

Selective responsivity to PAM treatment in mesenchymal cell variants was also seen in 3D bladder cancer cell line spheroids treated for 6 days. Activated caspase-3/7 staining was used to monitor apoptosis in spheroids, and Hoechst 33342 dye was used to counterstain the DNA (Figure 6a,b). PAM treatment (50% 10PAM gently mixed with the spheroids) induced apoptosis in all cell lines, with the TSU-Pr1-B2 showing less response than the -B1 or TSU-Pr1 cells (Figure 6a,b; representative spheroid pictures with green staining). Morphology of the spheroids became apoptotic, consistent with caspase-3/7 staining, and spheroids had a consistently lower area when measured in random fields after treatment with either cisplatin (standard-of-care chemotherapy for MIBC) or PAM (Figure 6c). Although this was only a single pilot experiment, the association of PAM response with epithelial/mesenchymal nature appeared to be consistent with the 2D results as mentioned above. 

After PAM treatment, the spheroid area of 5637 was reduced to 0.5 ± 0.2 times the original value, while that of TSU-Pr1 became 0.6 ± 0.2 times the original, TSU-Pr1-B1 became 0.5 ± 0.1 times, and TSU-Pr-B2 became 0.7 ± 0.1 times (Figure 6e). However, the use of 50% 10PAM, necessitated by not being able to readily remove the 3D media, was not as effective as seen with 70% 10PAM in 2D models. The reduced efficacy could also be caused by poor penetration, altered cell uptake, altered cell response in terms of ROS production, and/or increased antioxidant capacity within the cells in 3D. Further experimentation is required to accurately compare dose-responsiveness in 2D and 3D. The 3D culture treatment with 50% 10PAM was tested by an additional measure of cell viability, ATP production. The two different ways of measuring cell viability in the 3D setting provided comparable results and confirmed that the epithelial cell lines are more resistant to PAM treatment (Figure 6d,f).

### 3.7. The 3D Bladder Spheroid Model Confirms the Association of the Mesenchymal State and Cell ROS Level

We also assessed ROS staining of the PAM-treated 3D bladder cancer cell line spheroids. As seen in the representative images (Figure 7a), the ROS levels in TSU-Pr1-B2 spheroids were much lower than those observed in the other cell lines. ROS was mainly evident on the edge of the spheroids, rather than the spheroid core, which is consistent with the caspase-3/7 dye results demonstrating that only the edge cells were undergoing apoptosis. It seems likely that PAM penetration and its effects on ROS uptake/induction and apoptosis are limited to the cells on the spheroid periphery, although phenomena such as altered ROS half-life or altered induction of ROS production are possible (Figure 7b).

## 4. Discussion

Our observations support a close relationship between the mesenchymal state and the PAM response, which was confirmed by significant associations with EMT score using the 3 different algorithms (summarized in Figure 8). This is likely a consequence of elevated levels of ROS in these cell lines, as demonstrated in our study, but also reactive nitrogen species (collectively reactive oxygen and nitrogen species; RONS). ROS have been implicated in the selective killing of cancer cells by PAM. Cancer cells typically have higher resting levels of ROS than normal cells [38,39], and the additional RONS provided by PAM, raise these to toxic levels in cancer cells [1,3]. This is thought to be largely due to the inactivation of cell surface catalase, which otherwise eliminates cell-derived H_2_O_2_ [40]. Consistent with this, we found a strong correlation between the resting ROS level and cell response to PAM, with the highest ROS level observed in the mesenchymally-shifted cell lines. ROS levels have also been linked to mitochondrial dysfunction in mesenchymal stem cells from Parkinson’s disease [41] and linked to glucose metabolism in the EMT/breast cancer stem cell phenotype [42], as well as being linked to tumor recurrence/relapse [39]. Moreover, the ~3-fold increase in ROS levels after PAM treatment (Figure 2c) was higher in the mesenchymal cell lines than their respective epithelial counterparts, suggesting a facilitated uptake and/or reduced suppression of ROS in the mesenchymal lines that may relate to earlier associations of ROS with mitochondrial dysfunction in mesenchymal stem cells from Parkinson’s disease-affected individuals and with EMT traits in cancer stem cells [41,42]. We did see reasonable correlations between the EMT scores and the degree of induction of ROS across the cell lines analyzed. To our knowledge, this has not been reported before, and elucidation of the differential uptake induction of ROS is being explored with further studies.

In addition to the EMT score increases observed in the more mesenchymally-shifted cell lines, ROS values are also incrementally increased. These relatively high ROS values make the cells more sensitive to PAM treatment. After the PAM treatment, the majority of mesenchymally-shifted cells undergo apoptosis, while the remaining show distinct spindle-like, round, or polygon shapes. However, the epithelial cells are more resistant to PAM treatment, with only a fraction succumbing to PAM-induced apoptosis. The morphology of the surrounding cells remained consistent with that seen before treatment. These results are shown in both 2D monolayer and in our preliminary analysis of 3D spheroids. Only the edge cells of epithelial spheroids showed a response to PAM treatment, without any influence on the center cells. However, all the cells contained in spheroids generated from mesenchymally-shifted cancer cells responded to PAM treatment, resulting in apoptosis.

The enhanced responsiveness of mesenchymally-shifted cells to PAM, in each of the three models, strongly contrasts the observations that mesenchymally-shifted cancer cells are more resistant to various therapies [2,43]. The mesenchymal phenotype of low proliferation has been associated with cancer stemness, metastasis initiation, dormancy, and therapy resistance [13]. EMT has thus been well recognised as a potential target for oncotherapy [44], as EMT is more prevalent after treatment with various cancer therapies [45,46]. Thus, PAM and/or PAM-associated agents may provide an important avenue to target these cells that are constitutively aggressive and relatively resistant to current therapies.

Consistent with our findings, PAM was shown previously to be selectively active against TNBC cell lines [47], many of which are mesenchymal in nature (i.e., in the Basal A/BL1 or Basal B/M subgroups), although not all TNBC are more mesenchymal as shown by the EMT scores, PAM responses, and ROS levels of MDA-MB-468 cells, which are TNBC but more epithelial amongst the Basal A / BL1 subgroup. 

In the course of these studies, we discovered that the addition of 10% FBS to the PAM treatment reduced the effects of PAM on the cells, presumably due to a protection of the cells by the abundant protein. The repeat experiment performed alongside the BSA test showed that FBS had the opposite effect on MDA-MB-468 cells, where it slightly but significantly enhanced PAM activity (Appendix A), suggesting that these cells are more variable in their response and potentially sensitive to serum batch effect. However, an opposite effect was seen with 10% BSA, which exacerbated the cytotoxic effects of PAM. Direct activation of serum proteins by plasma resulted in higher activity (than PAM alone). Although the activation of FBS by plasma has been previously reported [48], little has been reported on how this affects cells or cell responses to PAM. The addition of serum (up to 30%) to PAM was shown to weaken the efficacy against glioblastoma cells, and it was proposed as a method to control PAM responses [37]. 

The effects of PAM on EMT were also sensitive to the presence of FBS; these effects were only seen in the presence of FBS and were most striking with the mesenchymal MDA-MB-231 cells, which are the most mesenchymal of the breast cancer cell lines. EMT reversal of PAM/FBS was very striking in these cells, with consistent changes in epithelial (upregulated) and mesenchymal (downregulated) gene expression. Some heterogeneous effects on both epithelial and mesenchymal markers were seen in MDA-MB-468 cells, potentially reflecting their known epithelial–mesenchymal plasticity [49,50]. Along similar lines, Adhikari et al. found that gas plasma combined with silymarin could downregulate mesenchymal markers *SNAI1* and *CDH2* and upregulate epithelial marker *CDH1* in metastatic melanoma cells [51]. Lee et al. showed that treatment of metastatic osteosarcoma cells with the CAP caused increased apoptosis and inhibited growth, along with migration and invasion being suppressed [9].

However, these studies were performed in 2D monolayer culture model. Although informative, such studies could not take the carcinoma microenvironment into consideration, which has been proven to play an essential role in response to treatments, especially in regulating tumor EMT processes [52]. Indeed, 3D models have emerged as a more highly representative culture format in cancer studies, particularly in relation to patient-derived tumoroids, which have the potential to recapitulate the patient response to specific agents [53,54]. Treatment of human glioblastoma spheroids with CAP treatment caused a reduction in the spheroids sizes and also inhibited the 3D migration [55]. Although only a pilot experiment, we also saw PAM could reduce the bladder cancer spheroids sizes and activate caspase-3/7 apoptotic signaling. 

There are a number of challenges to meet in translating PAM into clinical use, despite the incremental clinical scenarios for CAP/PAM testing [56,57]. Some examples include treatment of skin cancers [10] such as melanoma [58], as well as other surface-localized tumors [59]. Bladder tumors are amenable to PAM therapy as intravesical therapy is already a routine clinical practice [60]. Immunotherapies such as Bacillus Calmette-Guerin (BCG) treatment for NMIBC, and chemotherapeutic agents such as mitomycin and cisplatin, are delivered directly into the bladder when clinically indicated, which also helps avoid many of the side effects linked to systemic therapy. Thus, PAM therapy may be particularly relevant to organ-confined bladder cancers, particularly following transurethral resection of bladder tumor (TURBT) for MIBC.

By contrast, breast cancers are internal, and would require some new technological developments in terms of accessing PAM or CAP. Another possibility for treating breast cancers that are resected is to use PAM to flush the resection cavity to reduce the incidence of local recurrence. Importantly, EMT has been strongly linked to cancer stemness, particularly in breast cancer, where the breast cancer stem cell (CSC) profile shows commonality with EMT [13,15,61]. These cells are likely to seed local recurrences [62]. One of the traits of stemness is reduced proliferation, such that CSC can evade many of the proliferation-dependent therapies. Given the proliferation independent mechanism of action of PAM killing via ROS toxicity, PAM is likely to be effective against CSC which may be lying dormant outside of the surgical margins.

A minimally invasive approach for CAP delivery has been recently established, namely invivoPen, which can directly eject CAP in situ and is especially relevant for breast cancer treatment, which has the requirement for conservative surgery [63]. Alternatively, higher-grade lesions and metastatic sites could potentially be treated by intratumoral injection. We have shown that direct intratumoral injection of PAM could inhibit the growth of MDA-MB-231 xenografts [64], and image-guided intratumoral injection has become an accepted delivery route for immunotherapies in melanoma and other tumor types [65,66]. This includes anti-PD1 antibody (pembrolizumab ) delivered via the DfuseRx platform [67] or EBC-46 (tigilanol tiglate) [68]. Our results suggest that such technological developments and treatment variations may enable the use of an important novel treatment modality that selectively targets EMT-positive cells. Besides, the systemic injection of PAM or direct CAP treatment could expose tumour antigens to activate the immune system to target the escaped tumour cells, such process called immunogenic cell death [3]. Specific tumor-associated antigens released during the process have been observed to drive dendritic cells to maturity in vitro and subsequently activate T cells via secretion of cytokines [69].

## 5. Conclusions

In conclusion, we demonstrate an enhanced sensitivity to PAM in mesenchymally-shifted cancer cell populations in both breast and bladder cancer cell line systems characterized by epithelial–mesenchymal flux. Unlike many common therapies, PAM response is not dependent on the proliferative state of cells and, as such, is useful in mesenchymally-shifted cells or tumors. Further studies and technological developments are needed to actualize this potential, whereas our data strongly support the need for such studies.

## Figures and Tables

**Figure 1 cancers-13-02889-f001:**
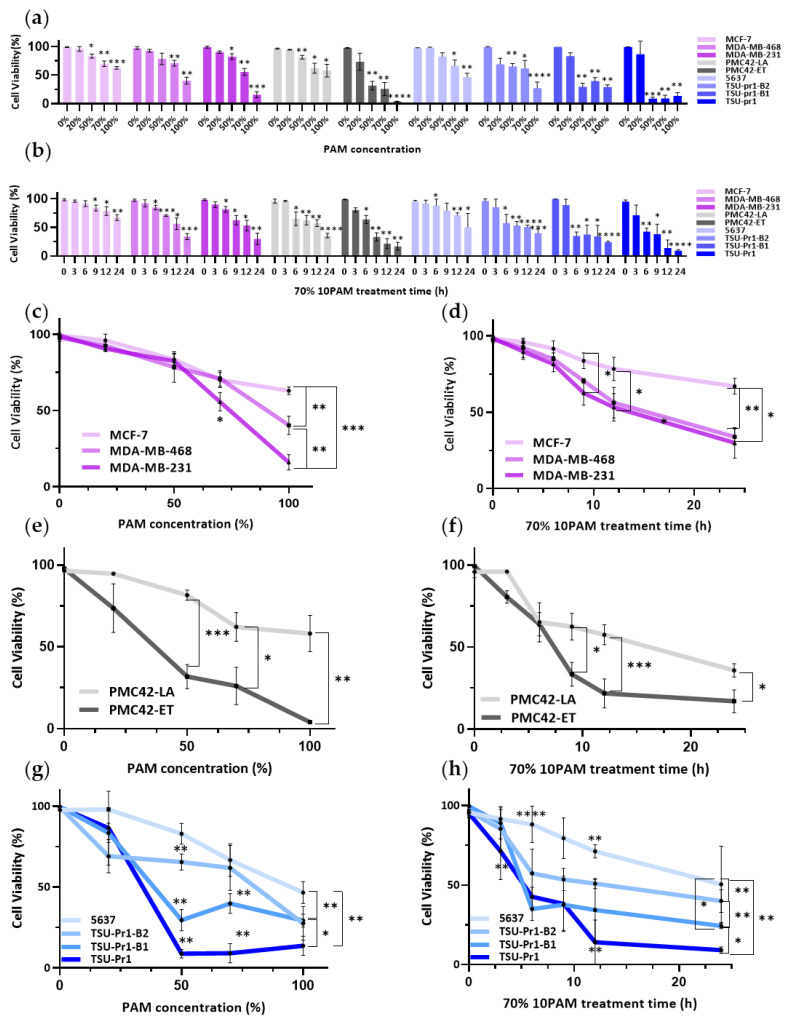
PAM selectively affects mesenchymally-shifted cancer cells. Breast cancer cell lines (MDA-MB-231/MDA-MB-468/MCF-7; pink) and EMT/MET cell line systems in breast (PMC42-ET/LA; grey) and bladder (5673/TSU-Pr1/B1/B2; blue) cancer were starved with serum-free medium for 3 h and treated with PAM in 0%, 20%, 50%, 70%, and 100% 10PAM for 12 h (**a**) or 70% 10PAM for 0, 3, 6, 9, 12, and 24 h (**b**). Cell assay viability as per live/dead cell assay (mean ± SEM of 0% or 0 h PAM, respectively) was calculated by IN Carta analysis. The dose response for 12 h treatment (**c**) and time course (**d**) of 70% 10PAM treatment of breast cancer cell lines (MDA-MB-231/MDA-MB-468/MCF-7; pink), replotted from (**a**,**b**), respectively; the dose response for 12 h treatment (**e**) and time course of 70% 10PAM treatment (**f**) of EMT/MET the PMC42 cell line system (ET/LA; grey); and the dose response for 12 h treatment (**g**) and time course of 70%10PAM treatment (**h**) of bladder cancer cells (5673/TSU-Pr1/B1/B2; blue) cells are shown. The mean and SEM of the average for each data point from each of the 3 experiments are shown. “*” represents *p* < 0.05, “**” represents *p* < 0.01, “***” represents *p* < 0.001, and “****” represents *p* < 0.0001.

**Figure 2 cancers-13-02889-f002:**
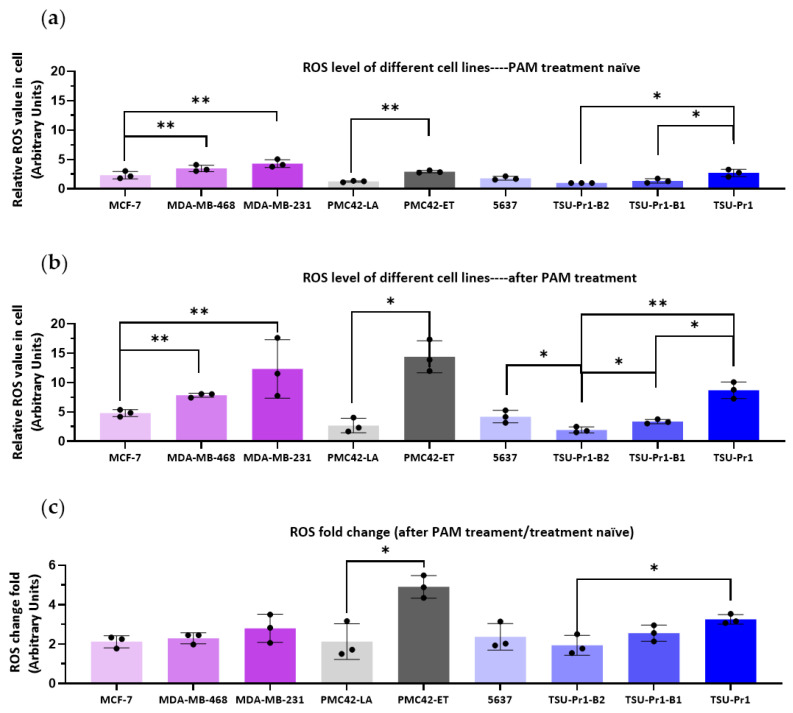
The effect of PAM on ROS levels in breast and bladder cancer cell lines. Breast cancer cell lines (MDA-MB-231, MDA-MB-468, MCF-7; pink) and EMT/MET cell line systems in breast (PMC42-ET/LA; grey) and bladder cancer (5673, TSU-Pr1/B1/B2; blue) were plated in a 96-well plate at 5000 cells per well overnight. After 3 h serum starvation, cells were incubated with CellROX Orange for 30 min in the incubator for ROS staining, then washed twice with PBS and stained with Hoechst 33342 for 15 min. The fluorescent images were taken by InCell 2000 and quantified by IN Carta. (**a**) The relative ROS level in the absence of PAM treatment compared to the level in TSU-Pr1-B2 cells. (**b**) ROS levels after 12h 70% 10PAM. (**c**) Fold change in ROS after 12 h 70% 10PAM (as per B) compared to resting state (as per A). (**d**) Fluorescent staining of ROS for each cell line in the resting state and after 12 h 70% 10PAM treatment, for the same cell lines as above (**a**). Nucleus is stained blue, ROS staining is yellow, and the merge of these is shown, as labeled. White arrows indicate dead or dying cells identified by disrupted nuclear morphology. The cell perimeters are outlined with white dashed lines. Bar = 5 μM. The mean and SEM of the average for each data point from each of the 3 experiments are shown. “*” represents *p* < 0.05, and “**” represents *p* < 0.01.

**Figure 3 cancers-13-02889-f003:**
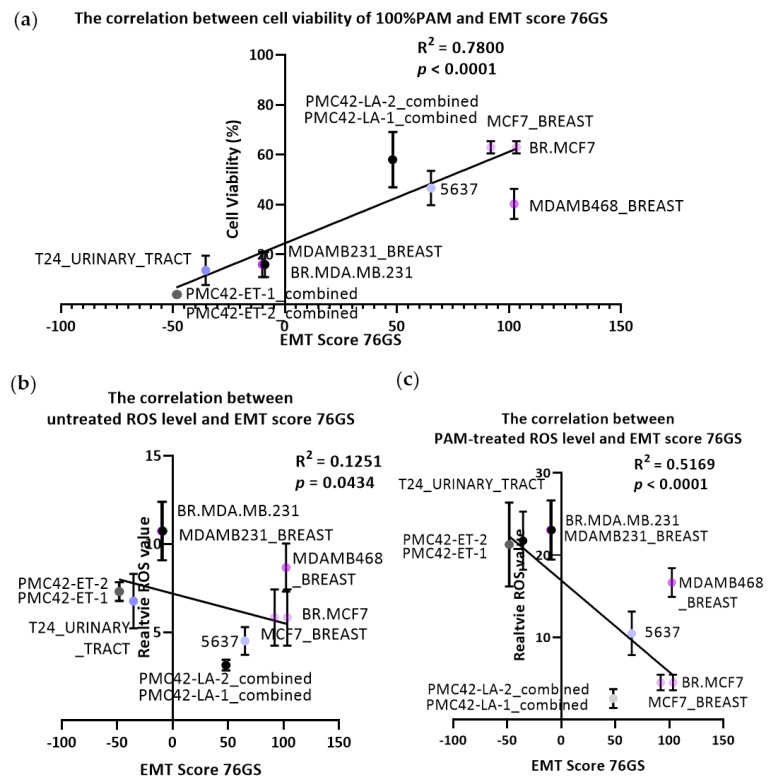
Transcriptomally derived EMT scores correlate with PAM response. (**a**) Correlation between cell viability following 12 h 100% 10PAM treatment and EMT score 76GS for MCF-7, MDA-MB-468, MDA-MB-231, PMC42-LA, PMC42-ET, 5637, and TSU-Pr1 (T24). In some cases, scores were derived using data for the same cell line from different public repositories (e.g., CCLE: BR.MCF7, BR.MDA.MB.231; NCI60: MCF7_BREAST, MDAMB231_BREAST). (**b**) Correlation between untreated (baseline) cellular ROS levels and EMT score 76GS. (**c**) Correlation between PAM-treated cellular ROS levels and EMT score 76GS.

**Figure 4 cancers-13-02889-f004:**
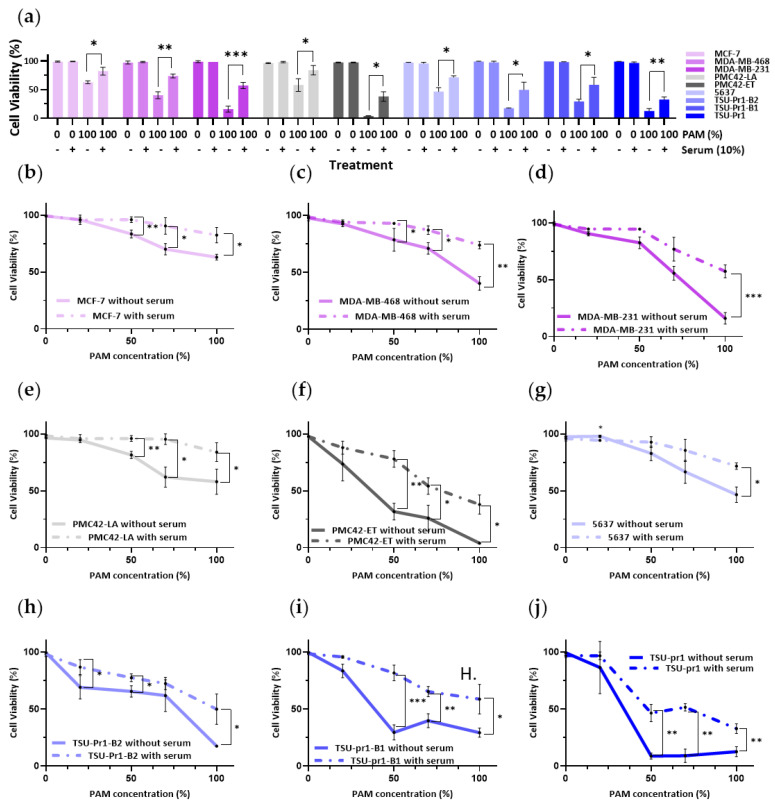
Impact of serum on PAM effect on cell viability. (**a**) Breast cancer cell lines (MDA-MB-231/MDA-MB-468/MCF-7; pink) and EMT/MET cell line systems in breast (PMC42-ET/LA; grey) and bladder (5673/TSU-Pr1/B1/B2; blue) cancer cells were plated in growth media for 24 h prior to 3 h serum starvation. Cells were then treated with 0% or 100% 10PAM in the absence or presence of FBS (10%) for another 12 h. After the treatment, these cells were stained with DAPI for DNA and PI for apoptosis assessment, and the cell viability was calculated by live/dead cell assay. MCF-7 (**b**), MDA-MB-468 (**c**), MDA-MB-231 (**d**), PMC42-LA (**e**), PMC42-ET (**f**), 5637 (**g**), TSU-Pr1-B2 (**h**), TSU-Pr1-B1 (**i**), and TSU-Pr1 (**j**) cells were plated as per (**a**) and treated with 0%, 25%, 50%, 75%, and 100% 10PAM for 12 h in the absence or presence of FBS (10%). The cell viability was tested as per (**a**); dotted lines represent plus serum. The mean and SEM of the average for each data point from each of the 3 experiments are shown. “*” represents a *p* < 0.05, “**” represents a *p* < 0.01, and “***” represents a *p* < 0.001.

**Figure 5 cancers-13-02889-f005:**
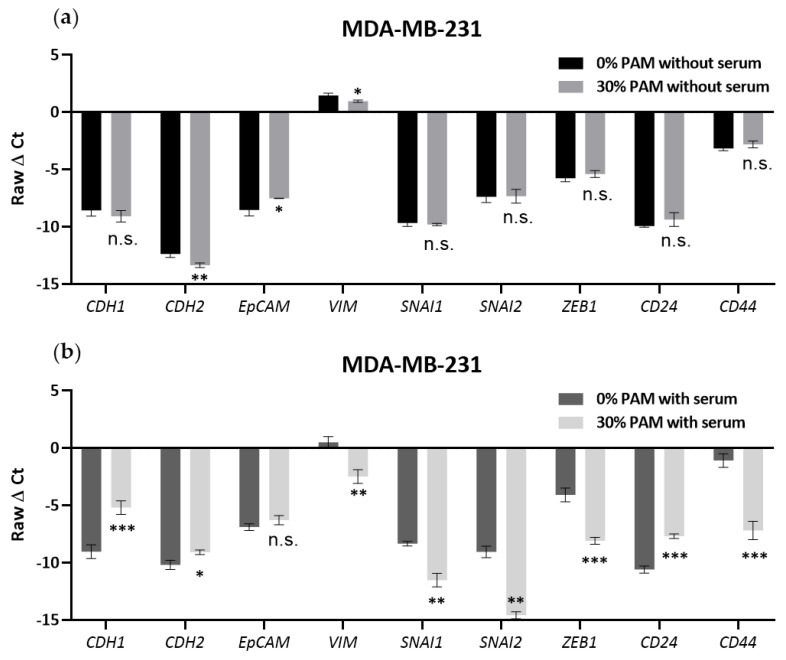
Changes in EMT markers after PAM treatment without or with serum. MDA-MB-231 cells were plated in a 48-well plate overnight and treated with 0% or 30% 10PAM without (**a**) or with 10% serum (**b**) for 3 days. Epithelial (CDH1), mesenchymal (CDH2, EpCAM, VIM, SNAI1, SNAI2, ZEB1), and stemness (CD24, CD44) markers were detected via real-time RT-qPCR. The mean and SEM of the average for each data point from each of the 3 experiments are shown. “*” represents a *p* < 0.05, “**” represents a *p* < 0.01, “***” represents a *p* < 0.001, and “ns” represents not significant.

**Figure 6 cancers-13-02889-f006:**
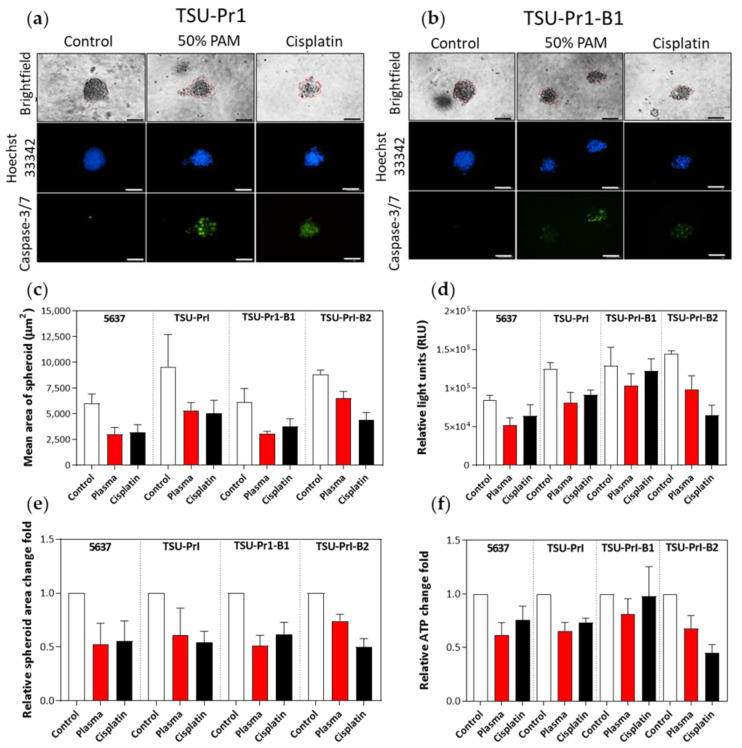
The 3D bladder cancer cells’ responses to PAM treatment. Representative images of spheroids from bladder cancer cell lines TSU-Pr1 (**a**) and TSU-Pr1-B1 (**b**). Treatment was performed for 6 days with control, 50% PAM, and 10 mM cisplatin. DNA was assessed with Hoechst 33342 (blue) and caspase-3/7 (green) staining using CellEvent Caspase-3/7 dye. Scale = 100 µm. (**c**) Area of spheroids from cell lines; control, treated with 50% PAM, or treated with 10 mM cisplatin (to mimic treatment of MIBC). SD calculated from the area of 5–10 spheroids. Only spheroids with ~>50 cells were counted. (**d**) Overall ATP production of treated bladder cancer cell line organoids was used as a measure of cellular viability, measured as RLU using the CellTiter-Glo 3D Cell Viability Assay following treatment. Average of 3 technical replicates normalized to background luminescence. (**e**) The relative fold change in spheroid area, representing the normalized data of (**c**). (**f**) The relative fold change in ATP levels, representing the normalized data of (**d**). These results are from a single pilot experiment with 10 spheroids for each treatment group.

**Figure 7 cancers-13-02889-f007:**
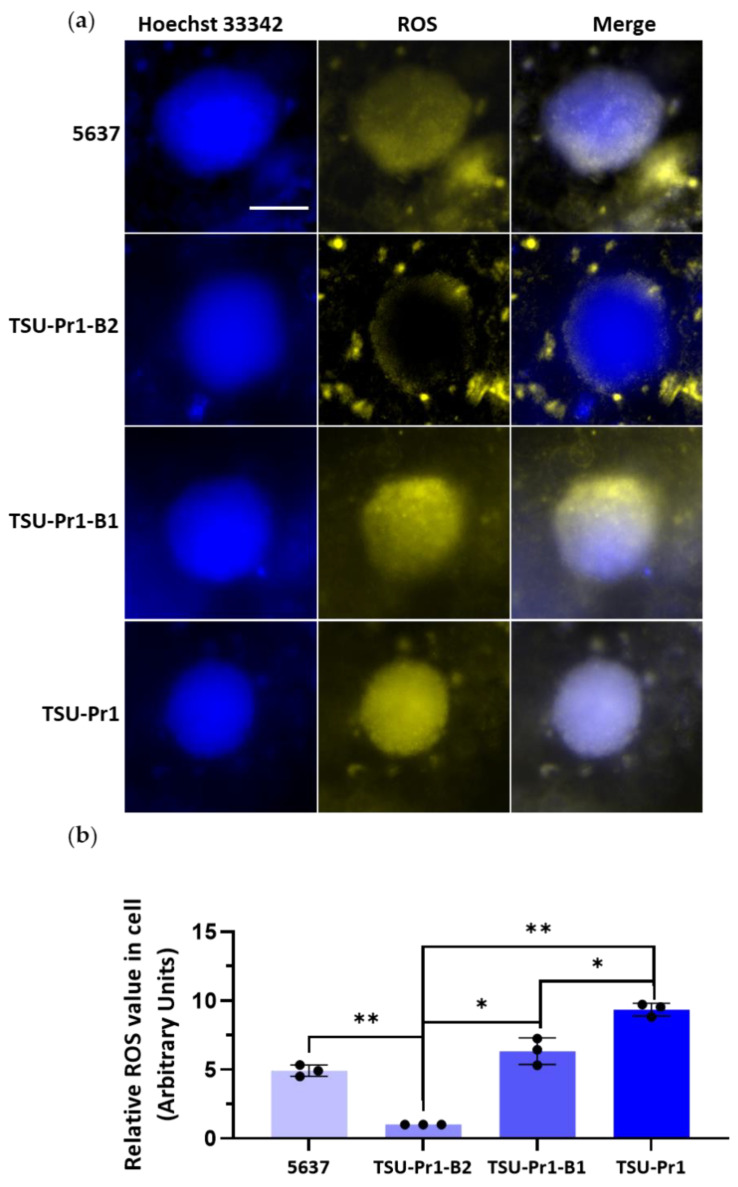
ROS level of PAM-treated 3D bladder cancer cell spheroids. (**a**) Representative images of ROS staining. After 6 days of 50% 10PAM treatment, the four cell line spheroids were stained for ROS and images were taken. DNA was assessed with Hoechst 33342 in blue and ROS staining with CellROX in yellow. The use of both of these stains in conjunction with the confocal capability of the InCell 6500HS instrumentation shows that the ROS staining is at the edge/surface, either surrounding the cell mass reflected by DNA staining (as in TSU-Pr1-B2) or above or below as in the other cell lines depicted. Scale = 100 µm. (**b**) The quantification of ROS level. Average of 3 technical replicates normalized to TSU-Pr1-B2 ROS luminescence. *N* = 1. These results are from a single pilot experiment with 10 spheroids for each treatment group. “*” represents a *p* < 0.05 and “**” represents a *p* < 0.01.

**Figure 8 cancers-13-02889-f008:**
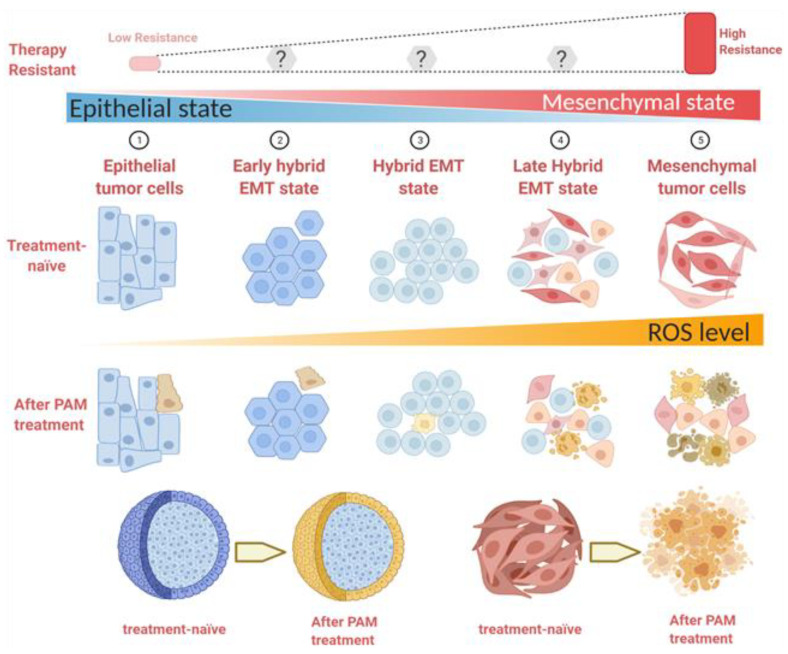
Schematic illustration of the relationship between EMT status, resistance to conventional therapies, responses to PAM, and ROS levels. Created with BioRender.com (accessed on 14 April 2021).

**Table 1 cancers-13-02889-t001:** The primer sequences of targeted genes.

Gene Symbol	Forward Primer Sequence	Reverse Primer Sequence
EpCAM	CTCGCGTTCGGGCTTCTGCTT	CAACTGAAGTACACTGGCATTGACGATTATT
SNAI2	CTGATGGCTAGATTGAGAGAATAAAAGACAGTAA	CACAGCAGCCAGATTCCTCATGTTT
SNAI1	CACATCCTTCTCACTGCCATGGAATT	CGCCTGGCACTGGTACTTCTT
VIM	CTAGAGATGGACAGGTTATCAACGAA	CCGTGAGGTCAGGCTTGGAAA
ZEB1	GTTACCAGGGAGGAGCAGTGAAA	GACAGCAGTGTCTTGTTGTTGTAGAAA
TWIST1	CTAGAGACTCTGGAGCTGGATAACTAAAAA	AAAGCTATTGATGGGCATGG
L32	CAGGGTTCGTAGAAGATTCAAGGG	CTTGGAGGAAACATTGTGAGCGATC
CD24	CTGTTCTCTTGGGAACTGAACTCACTTT	GTTGCCTCTC CTTCATCTTG TACATGAAA
CD44	GGTATCTCCTTTCTGAGGCTCCTACTAAAA	CTTCGACTGTTGACTGCAATGCAAA
CDH1	CTGTGCCCAGCCTCCATGTTTT	CAAGATGTGGCCAGACAAAGACACAAA
CDH2	CAGTAAAATTGAGCCTGAAGCCAACCTTA	AATGAAGATACCAGTTGGAGGCTGGTC

## Data Availability

The data presented in this study are available on request from the corresponding author.

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
