# Peer review of "Epithelial-to-Mesenchymal Transition Enhances Cancer Cell Sensitivity to Cytotoxic Effects of Cold Atmospheric Plasmas in Breast and Bladder Cancer Systems"

_cancers, 2021, doi:10.3390/cancers13122889_

Round 1
Reviewer 1 Report
comments are in attached .docx file

Reviewer 2 Report
This manuscript described the use of PAM in treating mesenchymal breast or bladder cancer cells with higher sensitivity. In general, this is an interesting observation, however, it is suggested that more detailed mechanisms should be examined before manuscript publication. Major parts:
1. Whether PAM or CAP treatment will fail under physiological serum concentration (100%) ? Again, whether 10% BSA, but not 10% FBS, also protect from CAM from cell damage?
2. The mechanisms of PAM in type-specific cells should be investigated. It was known that PAM might alter the intracellular signaling that resulted in reduced cell survival. Authors could examine whether mesenchymal breast or bladder cancer cells gained different levels of changes in specific signaling pathways.
3. As mentioned in the Discussion section, authors should investigate the correlation between the EMT markers and PAM treatments in breast or bladder cancer cells.
Minor parts:
1. Please make sure the images of ROS staining in TSU-Pr1-B1 cells in Fig.2d could represent the fold change shown in Fig.2c.
2. Several typos should be checked. For instance, page8, Line 243 "wek"?
3. In addition to cite references, the calculation of EMT scores should be described in Materials and Methods.
Reviewer 3 Report
Erik W. Thompson and colleagues have used cold atmospheric plasma (CAP) and plasma-activated medium (PAM) in context to epithelial-mesenchymal transition (EMT), metastatic potential, and resistance to conventional therapies in breast carcinoma and bladder cancer. In their study, authors have used breast cancer cell lines along with isogenic EMT/MET human breast cancer cells (PMC42-ET/LA) and observed more sensitivity to PAM treatment. Spheroid generated from bladder cancer cell lines displayed lower sensitivity to the PAM treatment. On the contrary, better responses were seen in mesenchymal cell lines. Authors have to address the following comments before this manuscript gets accepted
- Author should explain why they took 100%. In most of the cell lines, the cytotoxic effect is seen at 70% and 100%. I think these are very high percentages. What about the effect of these concentrations on normal mammary or other normal cell lines? Did the authors try 48 and 72 hours for the treatment?
- If the effect of PAM is more at EMT phenotype, then the authors should use TGF-B to induce EMT followed by PAM treatment
- Authors must show that what is the effect of PAM treatment of EMT markers. I am not sure whether PAM suppressed the EMT phenotype or the cytotoxic effect is independent of EMT.
- In the result section 3.1, the author did just mention Figure-1 but did not mention subheadings like (1a, 1b……1h). This is very inconvenient for reviewers to guess what the author wants to talk about their results. The author should write the results of 1C to 1H.
- the heading 3.2, the authors have talked about the ROS, what is the background to look at ROS levels
- ROS levels seen in the mesenchymal cell lines/variants (ET/LA, p=0.0160; TSU-Pr1/B2; p=0.0284). author should mention the fold change
- The ROS levels after 70%10PAM 12h treatment increased in all cell lines (Figure 2B). In my opinion, it should be Figure 2b
- The cell morphologies of epithelial cell lines, like MCF-7, PMC42-LA, TSU-Pr1-B2 201, and TSU-Pr1-B1, were cuboidal and closely adherent to each other (Figure 2a and d). where to look for morphology and what is the relation of ROS with morphology?
- In contrast, the mesenchymal counterparts exhibited a spindle-like and spread-out morphology (Figure 2a and d), where to look for morphology, and what is the relation of ROS with morphology?
- To convincing prove that PAM treatment suppressed the cancer stem cells or the stemness of the cancer cells as shown in the spheroid model; the Author must have to perform xenograft experiments.
- What is the effect of the PAM on stemness makers
- what is the effect of the PAM in combination with cisplatin or another drug?
- Author should perform a cell cycle and apoptosis experiment on the spheroid.
- author can have a look at these article in context to ROS and can improve the discussion PMID: 28587975; PMID: 32358622
- what is the mechanisms of induction of ROS upon PAM treatment
Round 2
Reviewer 1 Report
The reviewer wishes to express their thanks to the authors for the changes made to the manuscript. The new version is much clearer and the data added have made the argument more solid.
My residual concern is still with Section 3.6. I am still unconvinced that the evidence from the pilot study on the spheroids is strong enough to confirm the results from the 2D studies. I would suggest to change "confirm" into "tentatively confirms" or "indicates" or "suggests".
Reviewer 2 Report
Authors answered most of my questions with new results.
Reviewer 3 Report
Authors should explain why they have added few authors at this stage. Manuscript can be accepted in present form.